# Bone-to-Implant Contact in Implants with Plasma-Treated Nanostructured Calcium-Incorporated Surface (XPEEDActive) Compared to Non-Plasma-Treated Implants (XPEED): A Human Histologic Study at 4 Weeks

**DOI:** 10.3390/ma17102331

**Published:** 2024-05-14

**Authors:** Christian Makary, Abdallah Menhall, Pierre Lahoud, Kyung Ran Yang, Kwang Bum Park, Dainius Razukevicius, Tonino Traini

**Affiliations:** 1Oral Surgery Department, Saint Joseph University, Beirut P.O. Box 1104-2020, Lebanon; christian.makary@usj.edu.lb (C.M.); abdallah.menhall@usj.edu.lb (A.M.); pierre.lahoud@usj.edu.lb (P.L.); 2Daegu Mir Dental Hospital, Daegu 41934, Republic of Korea; ddsykr@gmail.com; 3MegaGen Implant Co., Ltd., Daegu 42921, Republic of Korea; periopkb@imegagen.com; 4Faculty of Odontology, Lithuanian University of Health Sciences, LT-50161 Kaunas, Lithuania; dainius@drdainiusrazukevicius.com; 5Department of Innovative Technologies in Medicine & Dentistry, University “G. d’Annunzio” of Chieti Pescara, 66100 Chieti, Italy

**Keywords:** titanium implants, plasma surface treatment, osseointegration, implantology, bone-to-implant contact, surface modification, immediate loading

## Abstract

Titanium implants undergo an aging process through surface hydrocarbon deposition, resulting in decreased wettability and bioactivity. Plasma treatment was shown to significantly reduce surface hydrocarbons, thus improving implant hydrophilicity and enhancing the osseointegration process. This study investigates the effect of plasma surface treatment on bone-to-implant contact (BIC) of implants presenting a nanostructured calcium-incorporated surface (XPEED^®^). Following a Randomized Controlled Trial (RCT) design, patients undergoing implant surgery in the posterior maxilla received additional plasma-treated (n = 7) or -untreated (n = 5) 3.5 × 8 mm implants that were retrieved after a 4-week healing period for histological examination. Histomorphometric analysis showed that plasma-treated implants exhibited a 38.7% BIC rate compared to 22.4% of untreated implants (*p* = 0.002), indicating enhanced osseointegration potential. Histological images also revealed increased bone formation and active osteoblastic activity around plasma-treated implants when compared to untreated specimens. The findings suggest that plasma treatment improves surface hydrophilicity and biological response, facilitating early bone formation around titanium implants. This study underscores the importance of surface modifications in optimizing implant integration and supports the use of plasma treatment to enhance osseointegration, thereby improving clinical outcomes in implant dentistry and offering benefits for immediate and early loading protocols, particularly in soft bone conditions.

## 1. Introduction

Titanium (Ti) fixtures are currently a quintessential part of modern dentistry [1]. They offer restorative solutions that would have otherwise been impossible [2]. With the fast evolution of all medical fields and the rise of the digital era, there is a continuous need for better, faster, and easier treatment modalities [3]. The modern dental implant has received countless upgrades and enhancements in almost every aspect when compared to its original predecessor [4]. Be it through modulating implant surface, geometry, or site preparation, most studies in that area aim at achieving osseointegration in shorter periods with higher short- and long-term success rates [5]. Implant microtopography is one of the main elements in that equation [6]. Many types of implant surface treatments are currently available on the market, such as sandblasting, acid etching, anodization, discrete calcium-phosphate crystal deposition, and coating with biologic molecules [7]. Recently, a surface treatment (XPEED^®^) consisting of Ca ion incorporation into a sandblasted, large-grit, acid-etched (SLA) surface in the form of nanostructured calcium titanate coating was shown to significantly enhance bone deposition around implants at early healing stages when compared to SLA alone [8]. This led to a better understanding of early phases of bone healing around implants and to a clinically proposed 4-week loading protocol [7]. These aforementioned surface treatment techniques have focused on enhancing bone apposition and healing, and implants are manufactured and packaged following certain protocols for optimal performance [9]. However, another indissociable factor from surface topography is surface chemistry, these are postulated to be two major factors that affect the osseointegration capacity of Ti implants, and studies show that oxidized Ti materials may induce superior osteoconduction [10]. That being said, studies also showed that Ti bioactivity decreases over time as hydrocarbons progressively deposit on its surface [11]. The term “biologic aging” was introduced referring to the time-related degradation of Ti bioactivity and function compared to freshly prepared Ti surfaces [12]. Irrespective of roughness degree, implant surfaces undergo changes from their manufacture to their clinical use [13]. Many publications discussed surface status alteration over time, from initially superhydrophilic to increasingly hydrophobic, lowering implant wettability, which determines the initial events and the biological cascade at the implant/bone interface [14,15,16,17]. Ultraviolet (UV) light-induced superhydrophilicity of titanium dioxide (TiO_2_) was described in 1997 [18]. This enhancement of implant surface hydrophilicity is mainly attributable to a decrease in hydrocarbon contamination and the introduction of a denser oxide layer [19]. UV treatment of aged Ti increases surface wettability which in turn positively affects recipient site biological response and seems to restore and even enhance bioactivity [20]. Plasma treatment consists of ionized gas (air, oxygen, argon, or nitrogen) in a vacuum chamber that forms plasma and removes contaminants from pure titanium metal surfaces and increases hydrophilicity, and it has been used for that purpose for several decades and has been shown to increase cellular adhesion to polymeric materials [21]. Numerous studies reported the advantages of implant plasma treatment, such as significantly higher in vitro fibroblast adhesion and proliferation compared to non-treated surfaces [17,22] and increased cytokine and growth factor secretion, resulting in improved wound healing [23]. However, a large majority of available studies in that area have described in vitro results and very few have tested surface treatment in vivo. The main objective of the present study was to test the efficiency of a plasma surface treatment method on implants placed in the posterior maxilla of human subjects and retrieved at 4 weeks in a RCT study design, by histologically comparing the percentage of bone-to-implant contact (BIC) between non-treated XPEED^®^ surfaces and treated surfaces (XPEED Active). The study null hypotheses under test considered no statistically significant differences in the BIC ratio for XPEED^®^ and XPEED Active implants after four weeks of healing.

## 2. Materials and Methods

### 2.1. Study Design

This was an RCT study in which mini implants (XPEED vs. XPEED Active) were used for ethical and economic reasons. Seven patients with bilateral edentulous posterior maxillae received two additional implants in the maxillary tuberosity area. The side (left or right) of placement for XPEED Active mini implants was randomly chosen for each patient. All inserted mini implants (n = 14) had the same geometry and surface (XPEED^®^, MegaGen, Gyeongsan, Republic of Korea). The XPEED Active group (n = 7) received surface treatment immediately prior to their placement using a novel plasma device (PlasmaX^®^motion; Daegu, Republic of Korea) with a 50 s cycle for each implant. The recruitment strategy and inclusion/exclusion criteria were the same as described in a previous study by the same authors [8].

### 2.2. Surgical Procedure

All procedures were performed in accordance with the recommendations of the Declaration of Helsinki for investigations in human subjects, as revised in Fortaleza. Patients received detailed information about the intervention and signed an informed consent form. The study was approved by the Ethics Committee at the Saint Joseph University of Beirut, Lebanon (USJ-2018-56). Prior to surgery, each case was clinically assessed and then radiographically evaluated with cone beam computed tomography (CBCT). Prosthetically driven treatment planning was carried out, and patients were then treated for both sides on the same day by the same experienced surgeon (C.M.) at the Oral Surgery Department, Faculty of Dental Medicine, Saint Joseph University (Beirut, Lebanon), between February and March 2023. Ten minutes prior to surgery, a one minute chlorhexidine digluconate solution (0.2%) intraoral rinse was applied. Under local anesthesia, a crestal full thickness flap allowed access to a residual bony ridge where implant sites were prepared following manufacturer’s recommendations, and each sector received two regular implants and one additional 3.5 × 8.5 mm mini implant distal to the most distal implant (AnyRidge, MegaGen, Gyeongsan, Republic of Korea). Test implants underwent plasma treatment immediately prior to placement using the PlasmaX^®^motion device (Daegu, Republic of Korea) connected to DC power. Treatment consists of placing a holder and a fixture driver equipped with a dental implant on the holder seating part, a plastic chamber is then lowered over the system, and a vacuum pump located behind the holder seating part is activated to obtain a pressure lower than 10 torr, removing air and impurities from the implant surface before plasma activation. When placed inside the machine, dental implant is automatically connected with a high voltage power supply electrode to apply a high voltage of up to 3 kV following a dielectric barrier discharge (DBD) configuration in order to discharge plasma in vacuum conditions, and a 50 s cycle for each implant is then performed. In the pumping state, the initial vacuum formation stage removes impurities from the implant surface as the gradient force discharges them out of the tube by a vacuum pump. At low pressure of < 13 mbar, plasma can be generated by the high voltage electrode, stimulating implant surface to facilitate the removal of additional impurities. This machine functions with an atmospheric pressure environment inside the tube, and no gas is used in the process. After implant placement, cover screws were placed, and flap was sutured for a submerged healing protocol. Periapical radiographs were then performed using a Rinn^®^ (Dentsply, Weybridge, UK) film holder for accurate paralleling and reproducible follow-up radiographs. *Per os* analgesics were prescribed, as well as antibiotic coverage (amoxicillin 2 g/daily, or in case of allergy, clindamycin 600 mg/daily) for 7 days and oral rinses of 0.12% chlorhexidine gluconate for 15 days. Second-stage surgery was performed at 4 weeks on both sectors according to the protocol described in an earlier study [7]. During this intervention, a trephine drill of 6.5/5.5 mm was used to retrieve mini implants. After retrieval, all biopsy sites were checked for fenestration/dehiscence and bony wall integrity. Two biopsies from the XPEED group were damaged upon retrieval and were not included in the results. Final restorations were delivered 4 weeks after healing abutment connection. Patients were then followed-up every 4 months for periodontal and oral hygiene maintenance.

### 2.3. Histological Processing

A cold 5% glucose solution was used to rinse the biopsies to eliminate blood residuals while preserving osmolarity (278 mOsm/L), and they were then placed in a 10% phosphate-buffered formalin solution sealed container at pH 7.1 for a period of 1 to 2 weeks depending on size. Following this process, biopsies were submerged in increasing ethanol concentrations as follows: 70% ethanol for 1 week; 80% ethanol for 1 week; 90% ethanol for 1 week; and 100% ethanol for 1 week. Specimens were then dehydrated and pre-infiltrated in a 50% resin/alcohol solution (LR White, London Resin Co., Ltd., Aldermaston, UK) for 10 days. This was followed by resin incorporation (Technovit 7200 VLC, Kulzer, Wehrheim, Germany). After polymerization, 50 μm undecalcified cut sections were processed and milled down to about 30 μm by using the TT System (TMA2, Grottammare, Italy). Bone implant contact measurements were made on sections stained with Azure II and methylene blue or acid fuchsine. A bright field light microscope (BX 51, Olympus America, Inc., Melville, NY, USA) connected to a high-resolution digital camera (FinePix S2 Pro, Fuji Photo Film Co., Ltd., Minato-Ku, Japan) were used for specimen inspection. A histometric software package with image capturing capabilities (Image-Pro Plus 6.0, Media Cybernetics Inc, Bethesda, MD, USA) was used. For added precision, software calibration was performed for each experimental image using “Calibration Wizard”, by measuring the number of pixels between two selected points of a micro-meter scale. Pixel number conversion into microns was then carried out to obtain distance values. The proportion between the entire perimeter of implant surface surrounded by bony tissue and the length of implant surface in direct contact with bone was calculated to assess the bone-to-implant contact rate (BIC). All measurements were performed by the same experienced operator (T.T.).

### 2.4. Statistical Analysis

The results were checked before for normal distribution using the Kolmogorov–Smirnov test. Assessing the normal distribution (*p* = 0.075), the data were statistically inferred using the unpaired Student’s *t*-test for XPEED and XPEEDActive. The level of statistical significance was set at *p* < 0.05. Statistical analyses were performed using IBM SPSS Statistics v. 3.5 (IBM Corp, Armonk, NY, USA).

## 3. Results

### 3.1. Clinical Outcome

All implants (control and test) healed uneventfully, as did all biopsy sites, and no complications were noted during the whole course of this study. At the 1-year follow-up, all sites showed uneventful healing with no clinical or radiographic complications.

### 3.2. Histomorphometric Analysis

Two of the seven control implants (the XPEED group) were not included in the morphometric evaluation due to the bone detachment that occurred probably during the retrieval procedure. The bone implant contact rate (BIC mean ± SD) for XPEEDActive was 38.7% (±8.5), while that for XPEED implants was 22.4% (±1.3) (Table 1). The difference of 16.3% was statistically significant (*p* = 0.002) (Figure 1). After 4 weeks of healing, plasma-treated implant surfaces showed a large amount of newly formed bone in direct contact with implant surface (Figure 2). Moreover, in many bone areas in contact with implant surface, an active osteoblastic rim cell was present. This observation was related to a high level of osteoconduction (Figure 3). The non-treated implant surface showed newly formed bone near the implant surface with some bone structs in contact with the implant, showing a moderate level of osteoconduction (Figure 4).

## 4. Discussion

The purpose of this paper was to validate an implant surface treatment method using non-thermal plasma. Implant surface hydrophilicity is one of the main factors responsible for cell adhesion and eventual fixture integration [24]. Surface topography is a key element in that equation, and studies show that there exists a certain range of roughness perceived by the cell that allows it to assume an osteoblastic morphotype instead of flattening and spreading into a fibroblastic morphology [25]. Various surface roughening techniques have been applied to improve the performance of the original machined surface [26]. Sandblasted, acid-etched surfaces (SLA) have proven to be reliable and safe, with higher success rates and faster loading time when compared to other commercially available implant surfaces [27,28,29]. However, nanostructured calcium-titanate-coated implant surfaces (XPEED^®^) showed higher BIC values at 4- and 6-week intervals when compared to SLA and machined surfaces, and they also appeared to promote better bone formation around the implant very early on after placement [8]. Biological aging starts as soon as the implant surface comes in contact with air [14]. It is a well-documented phenomena pertaining to the degradation of Ti bioactivity and function over time [12]. Att et al. showed that the levels of protein adsorption and osteoblast attachment to titanium surfaces were inversely correlated with the amount of hydrocarbon which increased in an age-dependent manner, and extrapolating that with time, titanium surface bioactivity constantly decreased until eventually reaching a level of bio-inertia [30]. This process was shown to affect all Ti implant surface types, hindering surface performance with increased storage time [20]. A substantial decrease in the adsorption rate of albumin, fibronectin, and serum proteins relevant to cell cultures was described on Ti surfaces after 4 weeks of storage when compared to “fresh” surfaces, with values exceeding 50% for albumin [12]. Consequently, many authors understood the importance of removing these hydrocarbon deposits, or preventing their onset prior to implant placement, without affecting surface topography and hydrophilicity [10,18,31]. One of the methods developed to counter the implant-aging process is a chemically modified SLA surface, SLActive (SLActive, Institute Straumann AG, Basel, CH, Switzerland), where implants are conserved in nitrogen in a sealed glass tube with an isotonic NaCl solution to avoid air contact and drying and maintain a clean TiO^2^ layer, preserving surface hydrophilicity and promoting osseointegration [32]. This modification in surface treatment and storage was shown to increase the bone apposition rate and BIC at the early healing stages when compared to the non-modified SLA surface [33]. In a histologic study on pigs, Buser et al. described an additional reduction in the healing interval as peri-implant bone growth was significantly faster at 2- and 4-week post-op [34]. Lang et al. reported higher degrees of osseointegration in human core biopsies retrieved 14 and 28 days after implant placement for SLActive implants compared to SLA [35]. However, in the same study, bone resorptive and appositional events were very similar for 7- and 42-day biopsies, indicating that both surfaces ultimately reach comparable levels of osseointegration after a certain period, fact that was confirmed by other authors as well [32,33,35,36]. Many authors focused on other methods to counter this degradation, light-induced surface treatment [11,37,38]. UV light-induced superhydrophilicity of titanium dioxide was first described by Wang et al. in 1997 [18]. Clinicians also investigated the effect of plasma treatment of Ti surfaces for accelerated cell adhesion, in regard to organic and inorganic surface contaminant removal [17,23,39,40,41]. In a systematic review by Pesce et al., the treatment of Ti dental implant surfaces using plasma or even UV was described as an effective method for improving the osseointegration process [42]. However, plasma treatment requires much less exposure time and confers higher osteoblast attachment and proliferation rates and a significantly better viability compared to non-treated and UV-treated surfaces [19,37]. Since in vitro results only have limited validity, in vivo studies are necessary to determine if and to what extent these results have effects in the complexity of a biological system [43]. Tsujita et al., in an in vivo study on rats, compared the performance of plasma-treated and non-treated Ti screws using micro-CT and histological analyses; their results show a clear improvement in surface performance for the test group compared to the control, which was attributed to the superhydrophilicity conferred by plasma [44]. In the present study, a new plasma device was used, consisting of the plasma processing of designated implants at room temperature under low pressure, with the aim of improving hydrophilicity and biological responsivity [24]. In this protocol, test implants were subjected to plasma processing using a chairside device (PlasmaX^®^motion; Daegu, Republic of Korea) directly prior to their placement. The treatment cycle duration was 50 s, as per manufacturer’s settings. This relatively short processing period compared to other studies, especially when comparing it to UV treatments, and thanks to the machine designed to hold and stabilize the implant/implant holder system inside the chamber with a magnetized platform, allowed the seamless incorporation of plasma treatment protocol into implant surgery immediately before implant placement without requiring any additional surgery time as implant treatment was performed while control implants were being placed [42].

Many clinical evaluation methods have been developed for the objective assessment of osseointegration (resonance frequency analysis, reverse torque analysis) [45,46,47,48]. Radiographic 3D bone-to-implant contact evaluation has been documented as more reliable and more reproductible than conventional linear 2D methods [49,50]. Fractal Dimension and Texture Analysis of osseointegration was proposed as an alternative and a much less invasive protocol, but the literature validating this method is still very scarce [51,52]. Histomorphometric BIC measurement and calculation still seem to be the gold standard in this area [26,53,54], even when compared to the micro-CT evaluation [55]. In vivo animal histological studies have already shown very promising results regarding plasma treatment. Long et al., in a study on beagle dogs, stated that plasma-treated implants enhanced cell interaction with material and subsequent bone formation, and their results showed that functionalized SLA implant performance was comparable to that of SLActive^®^ implant at the early osseointegration stages, and evidently superior to that of the untreated SLA group [56]. As stated in a previous study by the authors, data concerning human implant biopsies are still scarce, and most of the available publications in that area are retrospective and are usually performed on damage or temporary orthodontic implants retrieved for reasons such as connection fracture [8]. Studies on implants retrieved at early stages are even less common; in fact, few publications on that topic retrieved implants earlier than 6 months [57,58]. That being said, Lang et al. reported BIC values of SLA and SLActive^®^ implants placed in human volunteers at either 7, 14, 28, or 42 days [35]. Their results showed a clear superiority of the SLActive surface at 2 and 4 weeks with BIC rates of 14.8% and 48.3%, respectively, but almost identical values at 1 and 6 weeks. In the aforementioned publication by the authors [8], these values (SLActive^®^ 4-week BIC: 48.3%) were discussed and compared to ours (XPEED^®^ 4-week BIC: 22.4%), and the difference in values was mainly attributed to the implant site, since the latter retrieved the biopsies from the posterior maxilla while the former operated in the posterior mandible where bone density is significantly greater and BIC is proportional to the bone type [55]. When comparing the present study’s results with our earlier findings in an XPEED^®^ vs. SLA study with the same protocol and design, mean 4-week XPEED^®^ BIC values were identical for both studies (22.4%). This consistency in results helped benchmark the XPEEDActive values to be clearly superior to those of untreated implants with the same type of surface, which, in turn, were significantly higher than those of SLA implants at the early healing stages. Comparative studies between SLA and SLActive^®^ have reported better performance of the latter at 4 weeks and comparable BIC values at 6 weeks [35]. Therefore, it can be cautiously extrapolated that plasma-treated XPEED^®^ surfaces or XPEEDActive should yield higher BIC at these early healing stages. These hypotheses raise an interesting prospect of comparison between the performance of XPEEDActive and SLActive placed in human subjects in comparable bone qualities, allowing a clearer perspective on immediate and early loading protocols. Also, a similar study design to the present one should allow for the objective assessment of these variables with minimal invasiveness, low patient morbidity, and complete clinical healing, yielding statistically significant results even with small sample sizes, given the randomized, split-mouth protocol. In the current studies, implants were all placed in the posterior maxillary tuberosity for patients requiring implant placement in the posterior maxilla, and all implants meant for retrieval were placed during the same surgery, and there was no need for a second intervention since the flap was already opened. Also, biopsy retrieval did not require additional surgery since implants meant for prosthetic rehabilitation were submerged, and a second-stage surgery was planned. Tuberosity model presents advantages when bone is present in this region, and this was assessed based on the preoperative CBCT; thus, biopsies could be taken without harming the ridge. All biopsy sites were assessed after 1 year as explained in methodology and showed complete clinical healing. Bone in the maxillary tuberosity is usually a very soft bone consistent with type 4 bone following the Misch classification [59]. Implant surfaces were therefore tested under unfavorable conditions, and this will better emphasize the importance of high BIC observed in the present study. However, the study sample size was relatively small, and more objective data would require a larger number of biopsies for more conclusive data. Also, many factors had to be thoroughly assessed to ensure the success and low patient morbidity of this research, mainly careful planning and CBCT assessment, surgical expertise, and meticulous histological evaluation.

## 5. Conclusions

Our study compared in vivo performance of plasma-treated and non-treated implant surfaces, with the purpose of linking the experimental potential of implant surface treatment with clinical reality. Given the scarcity of similar human studies on this subject, our results provided important evidence supporting the efficacy of plasma treatment in enhancing bone-to-implant contact (BIC) values and early bone formation, as reported for the tested XPEEDActive implants when compared to the control group. Microscopic examination of tested implants showed a large amount of newly formed bone in direct contact with implant surface, and in many bone areas in contact with implant surface, an active osteoblastic rim cell was present, indicating a high level of osteoconduction.

These findings are crucial in the quest for optimizing implant success rates, especially in scenarios necessitating immediate or early loading protocols. This is of particular importance in challenging, soft, type 4 bone conditions such as the posterior maxilla, where sufficient stability and consequent integration are difficult to attain.

However, despite the promising outcomes, our relatively small sample size underscores the need for further research with expanded cohorts to corroborate our findings and ensure their validity and reliability.

That being said, it can already be established that this kind of surface treatment provides some advantages to implant dentistry, the extent of which is yet to be confirmed by studies with larger sample sizes and more follow-up. Also, a comparison of XPEEDActive and SLActive^®^ surfaces using the same protocol could also lead to a deeper understanding of these two different surface treatment methods and their relevance.

## Figures and Tables

**Figure 1 materials-17-02331-f001:**
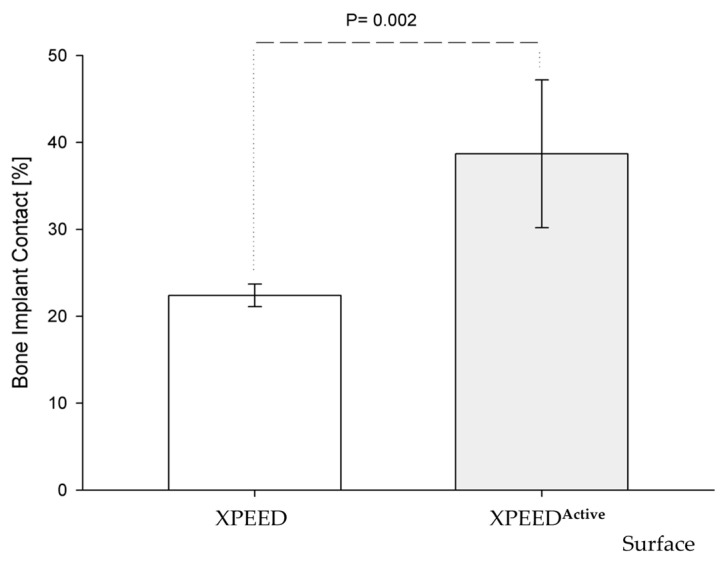
Graph of BIC rate vs. implant surfaces. The amount of bone around XPEED^Active^ implants was significantly more than XPEED implant (unpaired *t*-test, *p* = 0.002). 
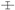
 standard deviation (SD).

**Figure 2 materials-17-02331-f002:**
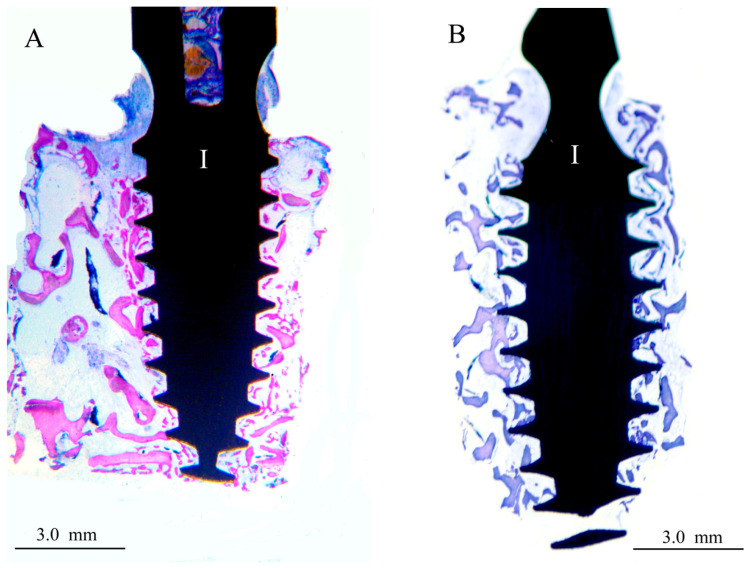
Images at 20× magnification. In the XPEEDActive sample (**A**), after 4 weeks of healing, a thin layer of bone appeared to have grown in direct contact with the implant surface. (I) implant, double-stain Azur II and fuchsine acid. In the XPEED sample (**B**), after 4 weeks of healing, bone growth appeared to be in direct contact with implant surface in some areas. (I) implant, double-stain Azur II and methylene blue.

**Figure 3 materials-17-02331-f003:**
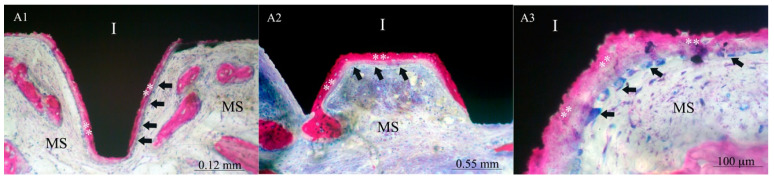
Images of XPEEDActive at 125× (**A1**), 70× (**A2**), and 1000× (**A3**). After four weeks, newly formed bone (**) appeared to grow mainly in direct contact with the implant surface, showing a high level of osteoconduction. In many areas, the newly formed bone (**) presents an active osteoblastic rim cell (black arrows). (MS) marrow spaces and (I) implant. Double stain with Azur II and fuchsine acid.

**Figure 4 materials-17-02331-f004:**
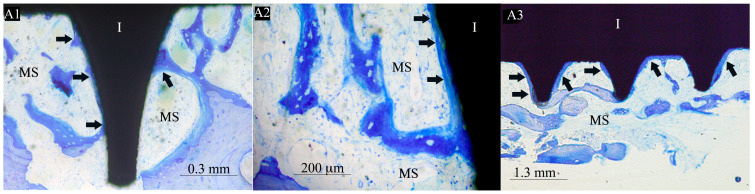
Images of XPEED at 50× (**A1**), 1000× (**A2**), and 25× (**A3**). After four weeks, the newly formed bone (black arrows) appeared to be present near the implant surface with some bone struts in contact with the implant, showing a moderate level of osteoconduction. (Black arrows) mineralized bone, (MS) marrow spaces, and (I) implant. Double stain with Azur II and methylene blue.

**Table 1 materials-17-02331-t001:** Statistical evaluation using unpaired *t*-test.

Group Name	N	Missing	Mean	Std Dev	SEM
XPEED ^1^	5	0	22.4	1.3	0.581
XPEEDActive ^1^	7	0	38.7	8.5	3.213
Mean Difference			−16.3		
t = −4.195 with 10 degrees of freedom (*p* = 0.002)					
95% confidence interval for difference of means: −24.957 to −7643					

^1^ The difference in the mean values of the two groups is greater than would be expected by chance; and there is a statistically significant difference between the input groups (*p* = 0.002). Power of performed test with alpha = 0.050:0.965.

## Data Availability

Data are available from the corresponding author upon request.

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
