# Peer review of "Bone-to-Implant Contact in Implants with Plasma-Treated Nanostructured Calcium-Incorporated Surface (XPEEDActive) Compared to Non-Plasma-Treated Implants (XPEED): A Human Histologic Study at 4 Weeks"

_materials, 2024, doi:10.3390/ma17102331_

Round 1

Reviewer 1 Report

Comments and Suggestions for Authors

The paper is generally well-structured with clear sections that logically flow from introduction to conclusions. The language is formal and appropriate for a scientific audience, with technical terms well-explained and consistently used.

However, few minor points should be addressed:

- The authors need to provide a short description of PlasmaX®motion; Daegu, Korea.

- More description for Figure 1.

- Add some color annotation for Fig 3.

- Also, compare your technologies with others: https://doi.org/10.1002/adfm.202310539; 10.1016/j.nano.2015.07.016

- Also, the authors may compare this technology with other coatings: https://doi.org/10.1002/adfm.202310539; 

Comments on the Quality of English Language

Few minor points in English need to be addressed.

Author Response

Firstly, we wish to express our sincere gratitude for the time and effort of the editorial team and the reviewers for the constructive suggestions and comments raised, that for sure were willing to help us in improving this manuscript. The answers for the comments of the reviewers and the editorial comments are reported  below point-by-point. As per journal indications, we also highlighted the relevant parts in main text for a clearer identification of the amendments.

The paper is generally well-structured with clear sections that logically flow from introduction to conclusions. The language is formal and appropriate for a scientific audience, with technical terms well-explained and consistently used. However, few minor points should be addressed:

Re:Thank you very much for the appreciation and for taking the time to review this manuscript. Please find the detailed responses below and the corresponding revisions/corrections highlighted in the main text.

Q: The authors need to provide a short description of PlasmaX®motion; Daegu, Korea.
Re:This was added to the text in line 115 as follows: “Test implants underwent plasma treatment immediately prior to placement using the PlasmaX®motion device (Daegu, Korea) connected to DC power. Treatment consists of placing a holder and a fixture driver equipped with a dental implant on holder seating part, a plastic chamber is then lowered over the system and a vacuum pump located behind the holder seating part is activated to obtain a pressure lower > 10 torr, removing air and impurities from implant surface before plasma activation. When placed inside the machine, dental implant is automatically connected with a high voltage power supply electrode apply a high voltage of up to 3kV following a dielectric barrier discharge (DBD) configuration in order to discharge plasma in vacuum conditions, and a 50-second cycle for each implant is then performed. In the pumping state, the initial vacuum formation stage removes impurities from implant surface due to gradient force discharges them out of the tube by vacuum pump. At low pressure < 13 mbar, plasma can be generated by the high voltage electrode, stimulating implant surface to facilitate the removal of additional impurities. This machine functions with atmospheric pressure environment inside the tube and no gas is used in the process.”

Q: More description for Figure 1.
Re: a more detailed description was added as follows:
“Graph of BIC rate vs surfaces, BIC mean (± SD) for XPEEDActive was 38.7% (± 8.5) while for XPEED implants was 22.4% (± 1.3).”

Q: Add some color annotation for Fig 3.

Re: color annotations were adjusted in  fig 3

Q: Also, compare your technologies with others:

 https://doi.org/10.1016/j.nano.2015.07.016

Re: the Editor vetoed the addition of these references

Q:Also, the authors may compare this technology with other coatings:

https://doi.org/10.1002/adfm.202310539

Re:the Editor vetoed the addition of these references

Reviewer 2 Report

Comments and Suggestions for Authors

In the manuscript titled „Bone to implant contact in implants with plasma-treated nanostructured Ca incorporated surface (XPEEDActive) compared to non-plasma-treated implants (XPEED): A human histologic study at 4 weeks.” The authors performed a study in which they implanted XPEED® and plasma treated XPEED mini implants in the maxillary tuberosity area in seven bilateral edentulous patients. The results state, that the experimental group has a significant higher bone to implant contact compared to the control group. The interpretation of the authors is that the higher osseointegration of the experimental group is favorable in immediate and early loading protocols as well as may improve the implant success rate.

Line 91 f you can’t choose something randomly. When you used a randomization please state how side where randomized and how many implants where placed on which side.

Line 95 “new plasma device” in case you mean newly bought than new is okay, when you mean it is recently developed than I suggest you use the word “novel” for “new”

Line 139 ff In regard to the Figure 3 and 4 the groups are stained differently if it is so, please specifically state that the samples are stained differently in regard to the groups and why that is done.

Line 153 ff since Student’s t-test were you please make statement about the normality of the data set. And since it is stated that there are only 7 patience and there are at least 7 experimental implants and the side were randomized it seems that follows that experimental and control implants were placed in the same individuals, therefore in my opinion a paired significant test need to be performed since the individuals have an influence on the bone regeneration.

General in Method part:

Usually, the amount of control and experimental implants is the same in your result table you only have 5 control but 7 experimental implants. Please be specific with how many mini implants you implanted in how many patients and the side since they were randomized. When only using 5 control with 7 experimental implants please state why you do not have the same amount of implants in each group. In case you excluded implants during the study please state why.

Figure 1. Since there are so few implants used, please change the bar chart to a scatterplot with boxplot that gives more and better information about the performance.

Figure 3 and 4 please unify the images in magnification, comparable area of the implant.

Discussion

There is a lot of just repetition of the introduction and listing of studies especially UV studies and this is not a study about UV activation. This is not helpful in a discussion.

In the beginning of the Discussion state very briefly the premise of the study and the main results and then interpret the results and put it in perspective of the recent knowledge and studies. This is not done by just by listing studies, but you have to point out the relation of your study with others similarities or differences and what the interpretation or conclusion is in this relation.

Line 346 “type 4 bone” please give a reverence for this kind of categorization.

Comments on the Quality of English Language

a few typos and formatting

Author Response

Firstly, we wish to express our sincere gratitude for the time and effort of the editorial team and the reviewers for the constructive suggestions and comments raised, that for sure were willing to help us in improving this manuscript. The answers for the comments of the reviewers and the editorial comments are reported  below point-by-point. As per journal indications, we also highlighted the relevant parts in main text for a clearer identification of the amendments.

In the manuscript titled „Bone to implant contact in implants with plasma-treated

nanostructured Ca incorporated surface (XPEEDActive) compared to non-plasma-treated

implants (XPEED): A human histologic study at 4 weeks.” The authors performed a study in

which they implanted XPEED® and plasma treated XPEED mini implants in the maxillary

tuberosity area in seven bilateral edentulous patients. The results state, that the experimental group has a significant higher bone to implant contact compared to the control group. The interpretation of the authors is that the higher osseointegration of the experimental group is favorable in immediate and early loading protocols as well as may improve the implant success rate.

Re:

Thank you very much for taking the time to review this manuscript. Please find the detailed responses below and the corresponding revisions/corrections highlighted in the main text.

Q: Line 91 f you can’t choose something randomly. When you used a randomization please state how side where randomized and how many implants where placed on which side.

Re:The number of implants and randomization aim were reported

Q: Line 95 “new plasma device” in case you mean newly bought than new is okay, when you

mean it is recently developed than I suggest you use the word “novel” for “new”

Re:the correction was introduced reporting novel instead of new

Q:Line 139 ff In regard to the Figure 3 and 4 the groups are stained differently if it is so, please specifically state that the samples are stained differently in regard to the groups and why that is done.

Re: both groups were double stained. The plasma treated group was stained with Azur II and fuchsine acid while the control group was stained with a metachromatic solution: Azur II and methylene blue (cationic dye). Both methods are known for the same affinity and no difference of sensitivity are reported (ref. Fritsch H. Staining of different tissues in thick epoxy resin-impregnated sections of human fetuses. Stain Technol. 1989;64(2):75-9. I used them to have a more pleasant contrast between the two groups improving the readability.

Q:Line 153 ff since Student’s t-test were you please make statement about the normality of the data set. And since it is stated that there are only 7 patience and there are at least 7

experimental implants and the side were randomized it seems that follows that experimental and control implants were placed in the same individuals, therefore in my opinion a paired significant test need to be performed since the individuals have an influence on the bone regeneration.

Re: The results were checked for normal distribution using the Kolmogorov-Smirnov test. Results showed a normal distribution (P= 0.075).  A paired t test was considered not appropriate since it is used in case the experiment subjects are observed before and after a single treatment. In fact, the statement of the hypothesis in paired t-test is to evaluate the average change the treatment produces instead of the difference in average responses with two treatments so, in our opinion, Unpaired t-test is more appropriate.

Q: General in Method part:

Usually, the amount of control and experimental implants is the same in your result table you only have 5 control but 7 experimental implants. Please be specific with how many mini implants you implanted in how many patients and the side since they were randomized. When only using 5 control with 7 experimental implants please state why you do not have the same amount of implants in each group. In case you excluded implants during the study. please state why.

Re: About the different numerosity we added an explanation in the text: Two of the seven control implants (XPEED group) were not included in the morphometric evaluation due to the partial  fracture and bone detachment from implant surface, occurred during the procedure.

Q: Figure 1. Since there are so few implants used, please change the bar chart to a scatterplot

with boxplot that gives more and better information about the performance.

Re: we prefer to use the bar chart instead of the boxplot since the difference in SD (1.3 vs 8.5) generate two box plot that appear to be much different and almost unreadable. While the data and the performance can be easily readable consulting the table 1 and the figures 2,3 and 4.

Q: Figure 3 and 4 please unify the images in magnification, comparable area of the implant.

Re: the magnification were accurately chosen to emphasize the results . On the other end, scale bars are reported for each one image to normalize the view.

Q: Discussion

There is a lot of just repetition of the introduction and listing of studies especially UV studies and this is not a study about UV activation. This is not helpful in a discussion.

In the beginning of the Discussion state very briefly the premise of the study and the main

results and then interpret the results and put it in perspective of the recent knowledge and

studies. This is not done by just by listing studies, but you have to point out the relation of

your study with others similarities or differences and what the interpretation or conclusion is in this relation.

Re: This was addressed in the discussion

Q: Line 346 “type 4 bone” please give a reverence for this kind of categorization.

Re: The reference was inserted as follows:

“Bone in the maxillary tuberosity is usually a very soft bone consistent with type 4 bone following the Misch classification(Misch CE. Bone classification, training keys to implant success. Dentistry today. 1989 May;8(4):39-44.)”

Reviewer 3 Report

Comments and Suggestions for Authors

This paper describes human histologic study on implants with plasma-treated nanostructured Ca incorporated surface (XPEEDActive) and non-plasma-treated implants (XPEED) as well as the outcome of experiments of four weeks’ period. The topic is really interesting, current and the paper contains many useful information and data on the present situation in practical use of these kind of implants. These data and the outcome results are very useful for scientists, dentists and patients alike. In general, this manuscript fills the gap in the scientific literature regarding the real applications of titanium implants and the potential outcomes by histologic study after four weeks.

The manuscript is well drafted, the methodology is clear and well described.

My comments and concerns regarding the content as follows:

The explanations and evaluations of Fig. 1-4 is missing. The Authors should describe what the pictures show in detail so that even nonprofessionals could understand.

The conclusion is consistent with the topic addressed, however, too concise and does not sufficiently highlight the final evidence and challenges of the application of the given implants and their final comparison in detail. I advise expanding the conclusions by going into more detail and emphasizing the most relevant achievements and difficulties.

The reference list does not reflect precisely the current state-of-the art in this particular field. So, more relevant references should be provided. The citations are mostly outdated.

Comments on the Quality of English Language

The quality of English Language is fine.

Author Response

Firstly, we wish to express our sincere gratitude for the time and effort of the editorial team and the reviewers for the constructive suggestions and comments raised, that for sure were willing to help us in improving this manuscript. The answers for the comments of the reviewers and the editorial comments are reported  below point-by-point. As per journal indications, we also highlighted the relevant parts in main text for a clearer identification of the amendments.

This paper describes human histologic study on implants with plasma-treated nanostructured Ca incorporated surface (XPEEDActive) and non-plasma-treated implants (XPEED) as well as the outcome of experiments of four weeks’ period. The topic is really interesting, current and the paper contains many useful information and data on the present situation in practical use of these kind of implants. These data and the outcome results are very useful for scientists, dentists and patients alike. In general, this manuscript fills the gap in the scientific literature regarding the real applications of titanium implants and the potential outcomes by histologic study after four weeks.

The manuscript is well drafted, the methodology is clear and well described.

My comments and concerns regarding the content as follows:

Re:

Thank you very much for taking the time to review this manuscript. Please find the detailed responses below and the corresponding revisions/corrections highlighted in the main text.

Q: The explanations and evaluations of Fig. 1-4 is missing. The Authors should describe what the pictures show in detail so that even nonprofessionals could understand.

Q: The conclusion is consistent with the topic addressed, however, too concise and does not sufficiently highlight the final evidence and challenges of the application of the given implants and their final comparison in detail. I advise expanding the conclusions by going into more detail and emphasizing the most relevant achievements and difficulties.

Re:

The conclusion was edited in the text as follows:

“Our study compared in vivo performance of plasma-treated and non-treated implant surfaces, with the purpose of linking the experimental potential of implant surface treatment with clinical reality. Given the scarcity of similar human studies on this subject, our results provided important evidence supporting the efficacy of plasma treatment in enhancing bone-to-implant contact (BIC) values and early bone formation, as reported for tested XPEEDActive implants when compared to control group. Microscopic examination of tested implants showed a large amount of newly formed bone in direct contact with implant surface, and in many bone areas in contact with implant surface, an active osteoblastic rim cell was present, indicating a high level of osteoconduction. These findings are crucial in the quest for optimizing implant success rates, especially in scenarios necessitating immediate or early loading protocols. This is of particular importance in challenging, soft, type 4 bone conditions such as the posterior maxilla, where sufficient stability and consequent integration are difficult to attain. That being said, it can already be established that this kind of surface treatment provides some advantages to implant dentistry, the extent of which is yet to be confirmed by studies with larger sample sizes and more follow-up. Also, a comparison of XPEEDActive and SLActive® surfaces using the same protocol could also lead to a deeper understanding of these two different surface treatment methods and their relevance.”

Q: The reference list does not reflect precisely the current state-of-the art in this particular field. So, more relevant references should be provided. The citations are mostly outdated.

Re:

Some references have been replaced

Reviewer 4 Report

Comments and Suggestions for Authors

This contribution studied the effect of plasma surface treatment on bone-to-implant contact (BIC) of implants presenting a nanostructured calcium-incorporated surface (XPEED®). The authors have considered that surface hydrocarbon deposition onto implants would decrease the surface wettability and bioactivity of the implants. Henceforth, the authors used the chairside plasma treatment immediately before the procedure could improve the implant surface hydrophilicity and enhance the osseointegration process.

This contribution described the rarely seen in vivo human trial data. It could be of great help for future large-scale clinical trials. Nevertheless, the authors should add more information regarding the plasma treatment parameters although the authors have mentioned this is a commercial instrument. These parameters include the pressure during the plasma treatment, what kind of gas is used, what power is used for generating the plasma, what kind of power is, DC or RF, and the treatment duration. Although the authors have mentioned the treatment duration in the Discussion section, it is better to have this information in the Materials and Methods section. All of the parameters mentioned have been studied to affect the plasma treatment efficacy in previous studies. Henceforth, the authors should mention these parameters clearly in the experimental procedure.

Author Response

Firstly, we wish to express our sincere gratitude for the time and effort of the editorial team and the reviewers for the constructive suggestions and comments raised, that for sure were willing to help us in improving this manuscript. The answers for the comments of the reviewers and the editorial comments are reported  below point-by-point. As per journal indications, we also highlighted the relevant parts in main text for a clearer identification of the amendments.

This contribution studied the effect of plasma surface treatment on bone-to-implant contact (BIC) of implants presenting a nanostructured calcium-incorporated surface (XPEED®). The authors have considered that surface hydrocarbon deposition onto implants would decrease the surface wettability and bioactivity of the implants. Henceforth, the authors used the chairside plasma treatment immediately before the procedure could improve the implant surface hydrophilicity and enhance the osseointegration process.

This contribution described the rarely seen in vivo human trial data. It could be of great help for future large-scale clinical trials. Nevertheless, the authors should add more information regarding the plasma treatment parameters although the authors have mentioned this is a commercial instrument. These parameters include the pressure during the plasma treatment, what kind of gas is used, what power is used for generating the plasma, what kind of power is, DC or RF, and the treatment duration.

Re:

Thank you very much for taking the time to review this manuscript. Please find the detailed responses below and the corresponding revisions/corrections highlighted in the main text.

Q: Although the authors have mentioned the treatment duration in the Discussion section, it is better to have this information in the Materials and Methods section. AM

Re:This has been added to the Materials and Methods section as follows:When placed inside the machine, dental implant is automatically connected with a high voltage power supply electrode apply a high voltage of up to 3kV following a dielectric barrier discharge (DBD) configuration in order to discharge plasma in vacuum conditions, and a 50-second cycle for each implant is then performed.

Q: All of the parameters mentioned have been studied to affect the plasma treatment efficacy in previous studies. Henceforth, the authors should mention these parameters clearly in the experimental procedure.

Re: Additional information about the used plasma treatment was incorporated in the Materials and Methods section as follows:

“Test implants underwent plasma treatment immediately prior to placement using the PlasmaX®motion device (Daegu, Korea)connected to DC power. Treatment consists of placing a holder and a fixture driver equipped with a dental implant on holder seating part, a plastic chamber is then lowered over the system and a vacuum pump located behind the holder seating part is activated to obtain a pressure lower > 10 torr, removing air and impurities from implant surface before plasma activation. When placed inside the machine, dental implant is automatically connected with a high voltage power supply electrode apply a high voltage of up to 3kV following a dielectric barrier discharge (DBD) configuration in order to discharge plasma in vacuum conditions, and a 50-second cycle for each implant is then performed. In the pumping state, the initial vacuum formation stage removes impurities from implant surface due to gradient force discharges them out of the tube by vacuum pump. At low pressure < 13 mbar, plasma can be generated by the high voltage electrode, stimulating implant surface to facilitate the removal of additional impurities. This machine functions with atmospheric pressure environment inside the tube and no gas is used in the process.”

Round 2

Reviewer 3 Report

Comments and Suggestions for Authors

The quality of the manuscript is acceptable after the revision. The answers are sufficient.